# First identification of genotypes of *Enterocytozoon bieneusi* (Microsporidia) among symptomatic and asymptomatic children in Mozambique

**Aly S. Muadica**[1], **Augusto E. Messa, Jr.**[2], **Alejandro Dashti**[1], **Sooria Balasegaram**[3], **Mónica Santin**[4], **Filomena Manjate**[2], **Percina Chirinda**[2], **Marcelino Garrine**[2,5], **Delfino Vubil**[2], **Sozinho Acácio**[2,6], **Pamela C. Köster**[1], **Begoña Bailo**[1], **Tacilta Nhampossa**[2,6], **Rafael Calero-Bernal**[7], **Jason M. Mwenda**[8], **Inácio Mandomando**[2,6]*, **David Carmena**[1]*

**1** Parasitology Reference and Research Laboratory, National Centre for Microbiology, Health Institute Carlos III, Majadahonda, Madrid, Spain, **2** Centro de Investigação em Saúde de Manhiça (CISM), Maputo, Mozambique, **3** Field Epidemiology Services, National Infection Service, Public Health England, London, United Kingdom, **4** Environmental Microbial and Food Safety Laboratory, Agricultural Research Service, United States Department of Agriculture, Beltsville, Maryland, United States of America, **5** Global Health and Tropical Medicine, Instituto de Higiene e Medicina Tropical, Universidade Nova de Lisboa (IHMT, UNL), Lisbon, Portugal, **6** Instituto Nacional de Saúde (INS), Ministério da Saúde, Maputo, Mozambique, **7** SALUVET, Department of Animal Health, Faculty of Veterinary, Complutense University of Madrid, Madrid, Spain, **8** Family and Reproductive Health Cluster, World Health Organization (WHO) Regional Office for Africa, Congo Brazaville

☯ These authors contributed equally to this work.
* inacio.mandomando@manhica.net (IM); dacarmena@isciii.es (DC)

**Data Availability Statement:** All relevant data are within the manuscript and its Supporting Information files.

## Abstract

*Enterocytozoon bieneusi* is a human pathogen with a broad range of animal hosts. Initially, *E. bieneusi* was considered an emerging opportunistic pathogen in immunocompromised, mainly HIV-infected patients, but it has been increasingly reported in apparently healthy individuals globally. As in other African countries, the molecular epidemiology of *E. bieneusi* in Mozambique remains completely unknown. Therefore, we undertook a study to investigate the occurrence and genetic diversity of *E. bieneusi* infections in children with gastrointestinal symptoms as well as in asymptomatic children in Mozambique. Individual stool specimens were collected from 1,247 children aged between 0 and 14 years-old living in urban and rural settings in Zambézia ($n = 1,097$) and Maputo ($n = 150$) provinces between 2016 and 2019. Samples were analysed for *E. bieneusi* by nested-PCR targeting the internal transcribed spacer (ITS) region of the rRNA gene. All positive amplicons were confirmed and genotyped. Penalised logistic regression (Firth) was used to evaluate risk associations. The overall prevalence of *E. bieneusi* in this children population was 0.7% (9/1,247). A 10-fold higher prevalence was found in Maputo (4.0%; 6/150) than in Zambézia (0.3%; 3/1,097). All *E. bieneusi*-positive samples were from children older than 1-year of age, and most (8/9) from asymptomatic children. Nucleotide sequence analysis of the ITS region revealed the presence of four genotypes, three previously reported (Peru11, $n = 1$; Type IV, $n = 2$, and S2, $n = 2$) and a novel genotype (named HhMzEb1, $n = 4$). Novel genotype HhMzEb1 was

**Funding:** DC received funding from the Health Institute Carlos III, Ministry of Economy and Competitiveness (Spain), under project PI16CIII/00024. TN received funding from the Fundo Nacional de Investigação, Ministry of Health (Mozambique) under project 245-INV. Additional funding was obtained from the USAID Country Office of Mozambique under the Fixed Amount Award Grant No. AID-656-F-16-00002 (IM) and the Centers for Disease Control and Prevention (CDC, USA) through the GAVI Alliance under subcontractor agreement MOA# 870-15 SC (JM). The funders did not play any role in the study design, data collection and analysis, decision to publish, or preparation of the manuscript.

**Competing interests:** The authors have declared that no competing interests exist.

identified in both asymptomatic (75%, 3/4) and symptomatic (25%, 1/4) children from a rural area in Maputo province in southern Mozambique. Genotypes HhMzEb1, Peru11, S2, and Type IV belonged to the Group 1 that includes genotypes with low host specificity and the potential for zoonotic and cross-species transmission. Being infected by enteric protozoan parasites and no handwashing were identified as risk associations for *E. bieneusi* infection. This study reports the first investigation of *E. bieneusi* genotypes in Mozambique with the identification of three previously reported genotypes in humans as well as a novel genotype (HhMzEb1). Findings highlight the need to conduct additional research to elucidate the epidemiology of *E. bieneusi* in the country, especially in rural areas where poor hygiene conditions still prevail. Special attention should be paid to the identification of suitable animal and environmental reservoirs of this parasite and to the characterization of transmission pathways.

## Author summary

*Enterocytozoon bieneusi* is an obligate intracellular parasite that infects a wide range of vertebrate hosts. It is the most important etiological agent of human microsporidiasis. Most clinical and epidemiological studies conducted to date have focused on immunodeficient or immunosuppressed individuals including HIV+ patients and solid organ transplant recipients, as in those *E. bieneusi* infection causes life-threatening chronic diarrhoea. In contrast, latent microsporidia infections in immunocompetent individuals have received far less attention. Molecular epidemiological studies in humans and animals have revealed that *E. bieneusi* encompasses a very large diversity of genetic variants (genotypes) with marked differences in host specificity and even geographical distribution. In Mozambique, as in many other African countries, the epidemiology of *E. bieneusi* is completely unknown. Therefore, to identify the occurrence and genetic diversity of this pathogen in Mozambique stool samples were obtained from children, including apparently healthy and symptomatic, in Zambézia and Maputo provinces and tested for *E. bieneusi* by molecular methods. Results demonstrated the presence of *E. bieneusi* genotypes for the first time in Mozambique. Four genotypes were identified, three genotypes that have previously been reported in humans (Peru11, Type IV, and S2) and a novel genotype (HhMzEb1). Two of the genotypes Peru11, Type IV have also been frequently identified in animals indicating that potentially zoonotic *E. bieneusi* genotypes are inadvertently circulating in the surveyed populations. Additional population genetic studies are needed to elucidate the actual extent of the epidemiology and transmission dynamics of *E. bieneusi* in Mozambique.

## Introduction

Microsporidia comprises 200 genera and nearly 1,500 species of spore-forming parasites ubiquitously found in the environment and able to colonize/infect a wide variety of invertebrate and vertebrate hosts [1,2]. Among the 17 Microsporidia species infecting humans *Enterocytozoon bieneusi* is the most frequently reported, causing gastrointestinal infections globally [3]. Infections with *E. bieneusi* in immunocompromised individuals (e.g. patients with AIDS, cancer, organ transplant recipients, and the elderly) are usually associated with chronic diarrhoea,

wasting syndrome, and weight loss. Infections in immunocompetent subjects are often asymptomatic or result in self-limited diarrhoea and malabsorption [4,5]. Extraintestinal disorders and pathologies have also been reported, mainly in lung [6]. In addition, *E. bieneusi* infection in early childhood has been recently linked to impaired growth in children in low-resource settings including African countries such as Malawi, South Africa, or Tanzania [7,8]. *Enterocytozoon bieneusi* has a high genetic diversity and is capable of colonizing/infecting a broad spectrum of mammal and avian species. There are marked differences in host specificities and zoonotic potential among those genotypes [9–11]. Infections are acquired through ingestion of food and water contaminated with *E. bieneusi* spores, or through direct contact with faeces of infected persons and animals or with contaminated soils [2,12,13].

The highly polymorphic ribosomal internal transcribed spacer (ITS) of the ribosomal rRNA (rRNA) gene is the most widely used marker for assessing the genetic diversity within *E. bieneusi* [11]. Based on ITS nucleotide sequences nearly 500 genotypes have been validated according to current nomenclature standards that are distributed into 11 distinct phylogenetic groups [10,14]. Groups 1 and 2 comprise genotypes able to infect a broad range of mammalian species including humans and are, therefore, considered potentially zoonotic. On the other hand, groups 3–11 have strong host specificities and are considered to pose little or null zoonotic risk [11].

The Global Enteric Multicenter Study (GEMS) has provided important information on the aetiology and population-based burden of paediatric diarrheal diseases in sub-Saharan Africa and South Asia. GEMS included in their study enteric protozoan parasites *Giardia duodenalis*, *Cryptosporidium* spp., and *Entamoeba histolytica*, but not the microsporidia *E. bieneusi* [15]. However, in recent years there has been a steady increase in the studies aiming to improve our understanding of the epidemiology of *E. bieneusi* in Africa since this parasite is frequently reported in immunocompromised individuals in African countries (Table 1 and S1 Fig). Importantly, early microscopy-based studies documented *E. bieneusi* prevalences as high as 40–67% in apparently healthy people in Cameroon [16] and Nigeria [17]. These findings indicate that asymptomatic, chronic *E. bieneusi* infections could be more frequent than anticipated. Despite the undoubted progress achieved over the past years, the actual epidemiological situation of *E. bieneusi* in most African countries, including Mozambique, remains largely unknown. To fill this gap, the present study aims to investigate the occurrence and genetic diversity of *E. bieneusi* in symptomatic and asymptomatic paediatric populations in two of the most populous provinces of Central (Zambézia) and Southern (Maputo) Mozambique.

## Methods

### Ethics statement

Written informed consent was obtained from legal guardians of children voluntarily participating in this survey. All procedures involved in the study were approved by the Ethics Committee of the Health Institute Carlos III under reference number CEI PI 17_2017-v3, and the National Bioethics Committee for Health (CNBS–Comité Nacional de Bioética para Saúde) of Mozambique under reference number 52/CNBS/2017 (participants in Zambézia). Similarly, approval by the Centro de Investigação em Saúde de Manhiça's Institutional Bioethics Committee for Health—CIBS (Ref. CIBS-CISM/075/2015), with final approval by CNBS (Ref. 209/CNBS/15), were obtained for participants in Maputo.

### Study design

This study is part of an ongoing research collaborative effort involving the Spanish National Centre for Microbiology (SNCM) in Spain, the Centro de Investigação em Saúde de Manhiça

**Table 1.** *Enterocytozoon bieneusi* infections reported in humans in African countries including country of origin, type of surveyed population, symptomatology, diagnostic method, and reported prevalence and genotypes. Diagnosis was conducted in stool samples except otherwise indicated.

| Country | Surveyed population | Clinical manifestations | Diagnostic method | Prevalence (%) | Genotype(s) | Reference |
|---|---|---|---|---|---|---|
| Cameroon | HIV+/TB+; HIV–/TB+; Immunocompetent | Diarrhoea | LM, IFAT | 35.7 (10/28); 24.0 (6/25); 67.5 (85/126) | ND | [16] |
| | HIV–; HIV+ | NS | IFAT, PCR | 2.9 (22/758); 0.5 (4/758) | A (8), B (3), CAF4 (5), D (3), Type IV[c] (1) | [36] |
| | HIV+ (adults) | Diarrhoea, asymptomatic | LM, PCR-RFLP | 6.5 (3/46); 4.6 (5/108) | Type IV (4) | [39] |
| | HIV– | None | PCR | 2.6 (5/196) | CAF1[d] (1), Type IV (4) | [51] |
| Chad | HIV+[a] | Diarrhoea | LM, IFAT, PCR | 100 (1/1) | ND | [52] |
| Democratic Republic of the Congo | HIV+ (> 15 yrs.) | Diarrhoea | PCR | 7.8 (19/242) | D (1), CAF1[d] (1), KIN-2 (1), KIN-3 (1), NIA1 (1) | [53] |
| | AIDS patients (> 15 yrs.) | Diarrhoea | IFAT, PCR | 5.1 (9/175) | NIA1 (2) | [54] |
| | AIDS patients | Diarrhoea, other | LM, IFAT, | 2.0 (1/50) | ND | [55] |
| Democratic Republic of São Tome and Principe | Rural children; Urban inpatient children | Diarrhoea, other | PCR | 5.2 (7/134); 8.9 (19/214) | CAF1[d] (4), KIN-3 (1), Type IV (14), A (2), D (5) | [30] |
| Ethiopia | HIV+, HIV– (adults) | Diarrhoea | U2B, PCR | 14.3 (15/105) | ND | [56] |
| | HIV+, HIV– (adults) | Diarrhoea, other | LM, PCR | 12.3 (30/243) | ND | [57] |
| Gabon | HIV+ (> 16 yrs.) | Diarrhoea | IFAT, PCR | 3.0 (25/822) | A (1), CAF1 (3), CAF2 (1), CAF3 (1), CAF4 (4), D (1), Type IV[c] (4) | [36] |
| Kenya | HIV+ (adults) | Diarrhoea | U2B | 3.0 (1/36) | ND | [58] |
| Madagascar | Inpatients (adults) | Diarrhoea, asymptomatic | PCR | 1.5 (1/67); 2.5 (5/198) | ND | [59] |
| Malawi | HIV+, HIV– | NS | PCR | 100 (37/37)[b] | D (4), Type IV[c] (9), Peru8 (1), S1 (1), S2 (11), S3 (2), S4 (1), S5 (4), S6 (2), UG2145 (1), | [35] |
| | HIV+ (< 18 yrs.) | Diarrhoea, other | qPCR | 37.0 (13/35) | ND | [7] |
| Mali | HIV+ (adults); HIV– (adults) | Diarrhoea | TEM, LM | 32.0 (28/88); 27.0 (3/11) | ND | [60] |
| | HIV+ Immunocompetent children | Diarrhoea | U2B, IFAT, PCR | 14.8 (9/61); 0.0 (0/71) | ND | [61] |
| Mozambique | Children and adults | Asymptomatic | PCR | 9.0 (27/301) | ND | [27] |
| | Children | Diarrhoea, asymptomatic | PCR | 0.3 (1/331); 0.9 (8/916) | HhMzEb1 (4), Peru11 (1), Type IV (2), S2 (2) | This study |
| Niger | HIV+ (Adults); HIV– (children) | Diarrhoea, other | TEM, LM | 7.0 (4/60); 0.8 (8/990) | ND | [62] |
| | HIV+ (mostly adults) | NS | LM, qPCR | 10.5 (24/228) | A (10), CAF1 (2), D (2), E (1), HAN1 (1) Type IV[c] (1), NIA1 (3) | [41] |
| Nigeria | Rural/urban dwellers | NS | LM | 39.6 (80/204) | ND | [17] |
| | HIV– (children) | Diarrhoea, asymptomatic | PCR | 9.3 (4/43) | D (2), Type IV[c] (1), novel K-like (1) | [31] |
| | HIV+, HIV– | Diarrhoea, other | PCR | 6.4 (10/157) | A/Type IV (1), D (3), D/WL7 (1), Type IV (1), WL7 (3), WL7/Type IV (1) | [42] |
| | HIV+ (adults) | Diarrhoea, other | PCR | 26.5 (26/98); 4.3 (52/365) | A (22), CAF2 (2), D (31), D/Type IV (1), EbpA (1), Type IV (14), Nig1(1), Nig2 (1), Nig3 (1), Nig4 (1), Nig5 (1), Peru8 (1) | [43] |
| | HIV+ | Diarrhoea | PCR | 7.7 (10/132) | D (1), Nig2 (2), Peru8 (1), Nig4-like (1), Type IV (5) | [63] |

(*Continued*)

**Table 1.** (Continued)

| Country | Surveyed population | Clinical manifestations | Diagnostic method | Prevalence (%) | Genotype(s) | Reference |
|---|---|---|---|---|---|---|
| | HIV+ (adults on HAART) | Diarrhoea, asymptomatic | PCR | 11.1 (12/108); 3.4 (6/177) | Mixed genotypes (3), Nig4 (2), Nig6 (10), Nig7 (2), Type IV (1) | [64] |
| | HIV+ | Diarrhoea, asymptomatic | LM, IFAT, PCR | 2.6 (5/193) | B (1), P-like, Peru3, PtEb IV, PtEb V, Type IV, UG2145 (2) | [65] |
| | HIV+ (adults) | Diarrhoea | PCR | 5.5 (5/90) | Type IV (4), unknown mixed infection (1) | [66] |
| South Africa | HIV+ (inpatients); HIV− (children) | Diarrhoea | LM, PCR-RFLP, qPCR | 12.9 (33/255); 4.5 (3/67) | ND | [32] |
| Tanzania | HIV+ (adults); HIV+, HIV−(children) | Diarrhoea, other; Chronic diarrhoea; Acute diarrhoea; Asymptomatic | LM, TEM | 3.5 (3/86); 3.4 (2/59); 0.0 (0/55); 20.0 (4/20) | ND | [34] |
| Tunisia | Immunocompromised patients | Diarrhoea | LM, PCR | 3.5 (3/86) | ND | [67] |
| | HIV+ (newborn) | Asymptomatic | LM, PCR | 100 (1/1) | ND | [68] |
| | HIV+ (adults); HIV− (adults) | Diarrhoea, other | LM, PCR | 20.0 (7/35); 5.3 (3/56) | ND | [69] |
| | HIV+ (adults) | Diarrhoea | LM, PCR | 2.5 (3/119); 5.9 (7/119) | ND | [70] |
| | HIV+ (mostly adults) | Diarrhoea | LM, PCR | 100% (7/7) | B (2), D (4), Peru8 (1) | [71] |
| | HIV+ (adults) | Diarrhoea, asymptomatic | LM, PCR | 19.4 (6/31); 2.8 (2/71) | ND | [72] |
| Uganda and Zambia | HIV+[b] | Diarrhoea, other | LM, TEM | 6.5 (5/77) | ND | [73] |
| | Children (< 5 yrs) | Diarrhoea, asymptomatic | LM, PCR | 17.4 (310/1779); 16.8 (112/667) | Type IV[c] (6), UG2145 (1) | [40] |
| | HIV+ (children); HIV− (children) | Diarrhoea, other | PCR | 76.9 (70/91); 6.6 (10/152) | ND | [33] |
| Zambia | Rural children | Asymptomatic | U2B, TEM | 0.6 (1/176) | ND | [74] |
| Zimbabwe | HIV+ (adults) | Diarrhoea | LM | 10.0 (13/129) | ND | [75] |
| | HIV+ (adults) | Diarrhoea | LM, PCR | 46.0 (34/74) | ND | [76] |
| | HIV+ (adults) | Diarrhoea | LM, PCR | 18.0 (10/55); 51.0 (28/55) | ND | [77] |
| | HIV− | Diarrhoea, other | LM, PCR | 33.0 (2/6) | ND | [78] |

HAART, Highly active antiretroviral therapy; HIV, Human immunodeficiency virus; IFAT, immunofluorescence antibody test; LM, light microscopy; ND, not determined; NS, not specified; PCR, polymerase chain reaction; qPCR, real-time PCR; RFLP, restriction fragment length polymorphism; TB, Tuberculosis; TEM, Transmission electron microscopy; U2B, Uvitex 2B fluorescent dye.

[a] Patient from Chad diagnosed in France.

[b] Diagnosed in an intestinal biopsy.

[c] Reported as genotype K.

[d] Reported as genotype KIN-1.

(CISM)/Fundação Manhiça (FM) in Mozambique, and the United States Department of Agriculture to assess the molecular epidemiology of the most frequent enteric protist parasites infecting humans in Mozambique. Two independent studies involving different sampling designs were conducted in Zambézia and Maputo provinces.

## Collection of stool samples in Zambézia province

Zambézia (population: 5.1 million; total area: 103,478 km$^2$) is the second most-populous province of Mozambique, located in the central coastal region of the country. A prospective cross-

sectional molecular epidemiological study was carried out between October 2017 and February 2019. All enrolled children were between 3 and 14 years old. A first set of samples were obtained from school children attending 18 public schools with each 35–2,111 (mean: 651) children in 10 of the 22 districts of Zambézia. Voluntary participants were provided with sampling kits (sterile polystyrene plastic flask with spatula and a unique identification number) to obtain individual stool samples. In addition, a second set of samples from children presenting with gastrointestinal complaints (abdominal pain, anal pruritus, bloating, constipation, diarrhoea, flatulence, loss of appetite, nausea, vomiting) seeking medical attention in health care centres (*n* = 6) and hospital settings (*n* = 1) from six districts of the province were collected.

For both sets, an aliquot of each faecal sample was transferred to a TOTAL-FIX stool collection device (Durviz, Valencia, Spain) and shipped to SNCM in Majadahonda (Spain) for downstream molecular analyses.

## Collection of stool samples in Maputo province

The study in the Maputo province was conducted in the Manhiça district (population: 0.2 million; total area: 2,373 km$^2$) that is located approximately 80 km north of the capital city Maputo in Southern Mozambique. The Manhiça district has been under a Health and Demographic Surveillance System (HDSS) from CISM since 1996, allowing the link between demographic and clinical data of the population [18].

Stool samples were collected from children under 5 years of age under an ongoing surveillance of diarrheal diseases conducted by the CISM since September 2015 in the Manhiça District Hospital (MDH) and other health peripheral facilities [18,19]. Children under 5 years of age presenting to the peripheral health facilities with moderate-to-severe and less-severe diarrhoea were recruited. Definition of diarrhoeal cases and inclusion criteria to participate in the research study were as previously described [20]. In addition, starting from December 2016, 1 to 3 community controls (asymptomatic children free of diarrhoea episode > 14 days) matched to the index case by age, sex and neighbourhood were identified (using the HDSS databases) and enrolled within 14 days after said index case was enrolled. Stools were collected for both cases and controls and sent to the laboratory for processing. Additionally, stools from children presenting to the sentinel health facility complaining with less-severe diarrhoea and those children presenting to the Xinavane Rural Hospital (XRH), were enrolled starting in April 2017 up to December 2018. Stool samples were collected in sterile flasks, transported to the laboratory, and stored at –80˚C without any preservatives for further testing of enteric pathogens as previously described [21].

## Data collection

Epidemiological and clinical information was collected in a standardized questionnaire provided as part of the sampling kits and labelled with the same identification number. The questionnaire was completed by researchers interviewing participating children and their caretakers at the time of sampling. The questionnaire included basic demographic characteristics (age, sex, place of living), and potential risk factors including contact with livestock or companion animals, type of drinking water, and information regarding the defecation place. There were some differences between Manhiça and Zambézia questionnaires.

## DNA extraction and purification

In stool samples from Zambézia, DNA was isolated from 200 mg of faecal material using the QIAamp Fast DNA stool mini Kit and the QIAcube robot (Qiagen, Hilden, Germany) following the manufacturer's instructions except that treatment of the samples with the lysis buffer

was carried out for 10 minutes at 95˚C. Purified DNA samples (200 μL) were stored at -20 ºC for downstream molecular analysis. In Manhiça, DNA was manually extracted and purified using the QIAamp DNA Stool Mini Kit from 200 mg of faecal material following the manufacturer's instructions, with the same modification for the lysing step. Purified DNA samples (200 μL) were shipped to the SNCM for molecular testing.

## Molecular detection of *Enterocytozoon bieneusi*

To detect *E. bieneusi*, a nested PCR was conducted to amplify a 390-bp fragment including the entire ITS as well as portions of the flanking large and small subunit of the rRNA gene [22]. Primary and secondary reactions (50 μL) were carried out using the outer primer set EBITS3 (5′–GGTCATAGGGATGAAGAG–3′) and EBITS4 (5′–TTCGAGTTCTTTCGCGCTC–3′), and the inner primer set EBITS1 (5′–GCTCTGAATATCTATGGCT–3′) and EBITS2.4 (5′–ATCGCCGACGGATCCAAGTG–3′), respectively. Reaction mixtures consisted of 1 μL of template DNA, 200 nM of each primer, 2.5 units of MyTAQ DNA polymerase (Bioline GmbH, Luckenwalde, Germany), and 5 μL of MyTAQ Reaction Buffer containing 5 mM dNTPs and 15 mM MgCl$_2$. Cycling conditions for the primary PCR were as follows: after denaturation at 94˚C for 3 min, samples were subjected to 35 cycles of amplification (denaturation at 94˚C for 30 s, annealing at 57˚C for 30 s, and elongation at 72˚C for 40 s), followed by a final extension at 72˚C for 10 min. Conditions for the secondary PCR were identical to the primary PCR except only 30 cycles were carried out with an annealing temperature of 55˚C. Negative (no DNA template) and positive (*E. bieneusi* PCR-positive samples, genotypes Ebcar2 and EbpA) controls were included in all PCR runs. PCR products were resolved on 2% D5 agarose gels (Conda, Madrid, Spain) stained with Pronasafe (Conda).

## Sequence analysis

All amplicons of the expected size were directly sequenced in both directions using the internal primer set in 10 μL reaction mixes using Big Dye chemistries and an ABI 3730xl sequencer analyser (Applied Biosystems, Foster City, CA). Sequencing data were viewed using Chromas Lite version 2.1 software (https://technelysium.com.au/wp/chromas/) to generate consensus sequences. The Blast tool (http://blast.ncbi.nlm.nih.gov/Blast.cgi) was used to compare those sequences with reference sequences deposited at the National Center for Biotechnology Information (NCBI). *Enterocytozoon bieneusi* genotypes were determined using the established nomenclature system based on ITS nucleotide sequence [14]. Sequences obtained in the present study were deposited in the GenBank database under accession numbers MN845065 to MN845068.

## Phylogenetic analysis

Nucleotide sequences obtained in this study and *E. bieneusi* nucleotide sequences for all genotypes previously identified in human and animals in Africa as well as appropriate reference sequences to include all *E. bieneusi* groups retrieved from GenBank were aligned with the Clustal W algorithm using MEGA X [23]. Phylogenetic inference was carried out by the Neighbor-Joining (NJ) method as previously described [24], genetic distance was calculated with the Kimura parameter-2 model using MEGA X [23].

## Statistical analysis

We analysed the data using EpiData 4.2.0 (EpiData Association, Odense, Denmark) and Stata software, versions 15 (STATA Corp., College Station, Texas, US). We calculated odds ratios

(OR) for associations; a probability (*P*) value < 0.05 was considered evidence of statistical significance. We examined for possible confounders (change of > 20% in OR) and interactions. As both populations included selection by symptomatic and asymptomatic populations, we included "a priori" the symptomatic variable in the multivariable model. As the event was rare, and some factors (e.g. *G. duodenalis*) were found in all cases, we used a penalised regression (Firth regression) selecting risk factors with a *P*-value ≤ 0.2 from the univariable analysis, using Akaike's information criterion (AIC) and Bayesian information criterion (BIC) to determine selection and evaluate the final model [25].

## Results

A total of 1,247 children aged between 0 and 14 years-old were recruited to participate in this study in the Manhiça District in Maputo (*n* = 150) and Zambézia province (*n* = 1,097) in Mozambique. The main socio-demographic and epidemiological features of these paediatric populations are summarized in Table 2.

**Table 2. Main socio-demographic and epidemiological features, expressed as frequencies, of the Mozambican children populations (*n* = 1,247) investigated in the present study.**

| | Zambézia | | | Manhiça district–Maputo | | |
|---|---|---|---|---|---|---|
| | Asymptomatic (*n* = 807) | Symptomatic (*n* = 290) | Total (*n* = 1,097) | Asymptomatic (*n* = 109) | Symptomatic (*n* = 41) | Total (*n* = 150) |
| **Gender** | | | | | | |
| Male | 49.7 | 50.3 | 49.9 | 54.1 | 48.8 | 52.7 |
| Female | 50.3 | 49.7 | 50.1 | 45.9 | 51.2 | 47.3 |
| **Age group (years)** | | | | | | |
| 0–5 | 13.4 | 33.8 | 18.8 | 100 | 100 | 100 |
| 6–10 | 64.3 | 49.3 | 60.3 | NA | NA | NA |
| 11–14 | 22.3 | 16.9 | 20.9 | NA | NA | NA |
| **Area** | | | | | | |
| Rural | 86.5 | 43.8 | 75.2 | 100 | 100 | 100 |
| Urban | 13.5 | 56.2 | 24.8 | NA | NA | NA |
| **Contact with livestock** | | | | | | |
| Yes | 4.6 | 14.1 | 7.1 | 59[b] | 46.3[c] | 55.3[b] |
| No | 95.4 | 85.9 | 92.9 | 41[b] | 53.7[c] | 44.7[b] |
| **Contact with dogs/cats** | | | | | | |
| Yes | 18.8 | 26.6 | 20.9 | 30[b] | 19.5[c] | 26.9[b] |
| No | 81.2 | 73.4 | 79.1 | 70[b] | 80.5[c] | 73.1[b] |
| **Main drinking water source** | | | | | | |
| River/springs | 4.3[a] | 4.1 | 4.3[a] | 2[b] | 2.4 | 3.6[b] |
| Tap | 23.2[a] | 54.1 | 31.4[a] | 91[b] | 80.5 | 86.5[b] |
| Well | 72.0[a] | 41.7 | 64.0[a] | 7[b] | 17.1 | 9.9[b] |
| **Defecation place** | | | | | | |
| Latrine | 88.0 | 76.9 | 85.1 | 100[c] | 100[d] | 100[e] |
| Outside | 12.0 | 23.1 | 14.9 | 0.0[c] | 0.0[d] | 0.0[e] |

NA: not applicable.

[a] Information for four children was not available.

[b] Information for nine children was not available.

[c] Information for one child was not available.

[d] Information for five children was not available.

[e] Information for six children was not available.

In Zambézia, 807 asymptomatic schoolchildren and 290 children presenting with gastrointestinal complaints seeking for medical care were investigated in 10 and six districts of the province, respectively (Table 2 and S1 and S2 Tables). The overall male/female ratio was 1.0, and children in the age group 6 to 10 years-old represented 60% of the surveyed individuals. Three out of four children investigated lived in rural areas.

In the rural Manhiça district, 109 asymptomatic children and 41 children presenting with gastrointestinal complaints (11 with moderate-to-severe diarrhoea and 30 with less severe diarrhoea) and seeking for medical care were investigated (Table 2 and S3 and S4 Tables). In this subpopulation, the male/female ratio was 1.1, and children in the age group 12–23 months-old represented 40.7% of the surveyed individuals. Most children (62.7%) were from the area covered by the MDH. All these children lived in rural areas.

Overall, *E. bieneusi* was detected by nested-PCR and confirmed by sequencing in 0.7% (9/1,247, 95% CI: 0.4–1.4) of the investigated children population. The prevalence was more than 10-fold higher in the Manhiça district in Maputo (4.0%, 6/150, 95% CI: 1.8–8.6) compared to all the districts in Zambézia combined (0.3%, 3/1,097, 95%CI: 0.1–0.8). The prevalence for those without symptoms was 0.9% (8/916, 95% CI: 0.4–1.7) and for those with symptoms 0.3% (1/331, 95% CI: 0.04–2.1), and those with diarrhoea 0.4% (1/251, 95% CI: 0.06–2.8). Nucleotide sequences of the ITS revealed the presence of four distinct *E. bieneusi* genotypes, three previously reported in humans (Type IV, S2, and Peru11) and a novel genotype (named HhMzEb1). No mixed infections were observed. Novel genotype HhMzEb1 differed from genotype D (AF101200) by one nucleotide. Genotype S2 nucleotide sequence obtained in this study showed a SNP at the large subunit region when compared with reference sequence (GenBank accession number FJ439678).

The main socio-demographic, epidemiological, and genotypes identified in each of the *E. bieneusi* infections identified in the present study are shown in Table 3. In the Zambézia province, *E. bieneusi* was found in 3 asymptomatic children. Two of them lived in a rural area, attended the same school, but were infected by different genotypes (Peru11 and S2) of the parasite. Both children were co-infected with *G. duodenalis*, *Blastocystis* sp., and *Strongyloides* spp. The third child was infected by the Type IV genotype of *E. bieneusi* and was co-infected with *G. duodenalis*. None of the symptomatic children investigated in this province tested positive to *E. bieneusi*.

In Maputo province, *E. bieneusi* was found in six male children, one presenting with moderate-to-severe diarrhoea and 5 asymptomatic. Five out of the six cases were younger than 2 years old. The S2 and Type IV genotypes were found in a single boy each, one of them co-infected with Rotavirus. Among the 4 boys infected with the novel genotype, HhMzEb1, one was HIV positive, presented with moderate-to-severe diarrhoea, and was co-infected with *Cryptosporidium meleagridis*. Another boy with this genotype was co-infected with *C. parvum*, and a third one was co-infected with Rotavirus. All six *E. bieneusi*-positive samples were co-infected with *G. duodenalis*.

Univariable analysis (Table 4) of the combined data showed that significant risk associations included infections with *G. duodenalis* or *Cryptosporidium* spp. and age, but handwashing was protective. In the Zambézia dataset alone, *G. duodenalis* was a significant association. In addition, travel, *Blastocystis* sp., and absence of a latrine were associated with presence of *E. bieneusi* (*P* value < 0.5), but confidence intervals crossed 1, with only 3 positive cases. In the Maputo dataset there were no significant associations. In the final multivariable model using the combined dataset, after adjusting for symptoms, the risk associations for *E. bieneusi* were infection with *G. duodenalis* [OR: 19.11, *P* = 0.43, 95% CI (1.10–332.94)]; infection with *Cryptosporidium* spp. [OR 5.75, *P* = 0.49 95% CI (1.01–32.89)] and handwashing [OR 0.31 *P* = 0.128 95%CI (0.67–1.41)]. Although handwashing was not significant, it was retained in

**Table 3. Main socio-demographic features, risk factors, and genotyping data of Mozambican children positive to *Enterocytozoon bieneusi* (*n* = 9) in the present study.**

| Province | District | Sample ID | Area | Gender | Age | Contact with livestock and/ or poultry | Contact with companion animals | Main source of drinking water | Defecation place | Genotype | GenBank accession number |
|---|---|---|---|---|---|---|---|---|---|---|---|
| Zambézia | Gurúe | 18 | Urban | Male | 132 months-old | No | Cat, dog | Public tap | Latrine | Type IV | MN845065 |
| | Mocuba | 207 | Rural | Male | 60 months-old | No | No | Well | Outside | Peru11 | MN845067 |
| | Mocuba | 210 | Rural | Female | 60 months-old | No | No | Well | Outside | S2 | MN845066 |
| Maputo | Manhiça–MDH | 1728661.9[a, c, d] | Rural | Male | 16 months-old | Poultry | No | Public tap | Latrine | HhMzEb1[b] | MN845068 |
| | Manhiça–MDH | 1725028.3 | Rural | Male | 28 months-old | ND | ND | ND | Latrine | HhMzEb1[b] | Identical to MN845068 |
| | Manhiça–XRH | 1754917.2[e] | Rural | Male | 16 months-old | ND | ND | ND | Latrine | HhMzEb1[b] | Identical to MN845068 |
| | Manhiça–XRH | 1725031.3[d, e] | Rural | Male | 12 months-old | No | No | Public tap | Latrine | Type IV | Identical to MN845065 |
| | Manhiça–XRH | 1754898.4 | Rural | Male | 12 months-old | Goat, poultry | No | Unprotected well | Latrine | S2 | Identical to MN845066 |
| | Manhiça–MDH | 1754369.9[f] | Rural | Male | 15 months-old | No | No | ND | Latrine | HhMzEb1[b] | Identical to MN845068 |

ND: no data available; MDH: Manhiça District Hospital; XRH: Xinavane Rural Hospital.

[a] Symptomatic child presenting with moderate-to-severe diarrhoea.

[b] Novel genotype.

[c] Co-infected with *Cryptosporidium meleagridis* and HIV positive.

[d] Deceased.

[e] Co-infected with Rotavirus.

[f] Co-infected with *Cryptosporidium parvum*.

the final model as it confounded the relationship with *Cryptosporidium* spp. Age was not a factor after adjustment for other variables.

Phylogenetic analysis revealed that the novel genotype HhMzEB1 clustered within Group 1 when its relationship with other genotypes of human and animal origin previously reported in Africa was evaluated (Fig 1).

## Discussion

Little information is currently available on the occurrence and distribution of enteric parasites in Mozambique [26–28], although the GEMS project has significantly improved our understanding of the epidemiology of diarrhoea-associated with parasites *G. duodenalis*, *Cryptosporidium* spp., and *E. histolytica* [15, 29]. However, there is almost no information regarding Microsporidia, and only one study that aimed to investigate the prevalence of intestinal parasite infections in an informal settlement in Beira, Mozambique has reported *E. bieneusi* [27].

**Table 4. Descriptive and univariable analysis of the variables of interest potentially associated with an increased exposure risk to *Enterocytozoon bieneusi* in the present study.**

| Variable | Cases of *E. bieneusi* with variable | % | Non cases with variable | % | Odds ratio | 95% CI | P value |
|---|---|---|---|---|---|---|---|
| *Cryptosporidium* spp. | 2 | 22 | 35 | 3 | 9.82 | 0.96–53.84 | 0.001 |
| *Giardia duodenalis* | 9 | 100 | 556 | 45 | 15.27 | 2.87–. | 0.001 |
| Age | Continuous | | | | 0.66 | 0.51–0.85 | 0.001 |
| Handwashing (yes) | 6 | 67 | 1152 | 93 | 0.14 | 0.03–0.86 | 0.001 |
| Male Sex | 7 | 78 | 622 | 50 | 3.47 | 0.66–34.29 | 0.1 |
| Has livestock | 2 | 22 | 150 | 12 | 3.61 | 0.32–25.37 | 0.115 |
| Water source (not tap water) | 3 | 33 | 885 | 71 | 0.39 | 0.05–2.94 | 0.236 |
| Symptomatic | 1 | 11 | 330 | 27 | 0.34 | 0.01–2.58 | 0.293 |
| *Blastocystis* sp. | 2 | 22 | 156 | 13 | 1.98 | 0.20–10.51 | 0.387 |
| No latrine | 2 | 22 | 162 | 13 | 1.89 | 0.19–10.01 | 0.423 |
| Rural | 8 | 89 | 967 | 78 | 2.24 | 0.30–99.83 | 0.435 |
| Has pets (dog/cats) | 1 | 11 | 267 | 22 | 0.72 | 0.02–6.50 | 0.766 |
| *Stronglyoides stercoralis* | 2 | 22 | 253 | 20 | 1.11 | 0.11–5.88 | 0.895 |
| *Entamoeba dispar* | 1 | 11 | 122 | 10 | 1.14 | 0.03–8.65 | 0.9 |

95% CI: 95% Confidence Interval.

They detected *E. bieneusi* by real-time PCR in 9.3% (28/301) of the individuals tested and no genotyping was provided. Therefore, this is to our knowledge the first molecular epidemiological study describing occurrence and genetic diversity of *E. bieneusi* in Mozambique. The overall prevalence of *E. bieneusi* infection in the paediatric (0–14 years of age) populations from Mozambique tested in this study was 0.7% (9/1,247). Infections were more commonly identified in asymptomatic children in rural settings.

The *E. bieneusi* infection rates found in our paediatric populations in Zambézia (0.3%) and Maputo (4%) were lower than the prevalence reported in Beira in Sofala province (9%) [27], indicating potential differences in the geographical distribution of the parasite in Mozambique. The disparity could also be associated with differences in socioeconomic characteristics of the studied population as the study in Beira was conducted in an informal settlement in an area that is frequently flooded and not connected to a sewage system after a local hospital noticed high number of diarrhoea cases in this settlement. Prevalence in Mozambique (0.7%) is lower to those previously documented in immunocompetent children populations (range: 4–9%) in other African countries including the Democratic Republic of São Tomé and Principe [30], Nigeria [31], South Africa [32], and Uganda [33], and much lower than those found in similar paediatric populations (range: 20–67%) in Cameroon [16], Nigeria [17], and Tanzania [34] (see Table 1). However, prevalence is influenced by the age group selected in the study and the presence of symptoms, so comparisons among studies must keep those differences in mind.

It is noteworthy to indicate that all *E. bieneusi*-positive children were also co-infected by other pathogens including two children in Manhiça that were HIV-positive. Indeed, one to three additional enteric parasites (*G. duodenalis*, *Cryptosporidium* spp., *Blastocystis* sp., and *Strongyloides* spp.) were found and the strongest risk associations were with *G. duodenalis* and *Cryptosporidium* spp. These data clearly depict a highly endemic scenario where polyparasitism is common as previously reported in the city of Beira [27].

Our study is limited by the small number of positive samples, thus the wide confidence intervals for our results. Furthermore, detailed information on handwashing (use of soap or not) and latrine use (toilet or outside latrine) were unavailable in the Zambézia dataset, and

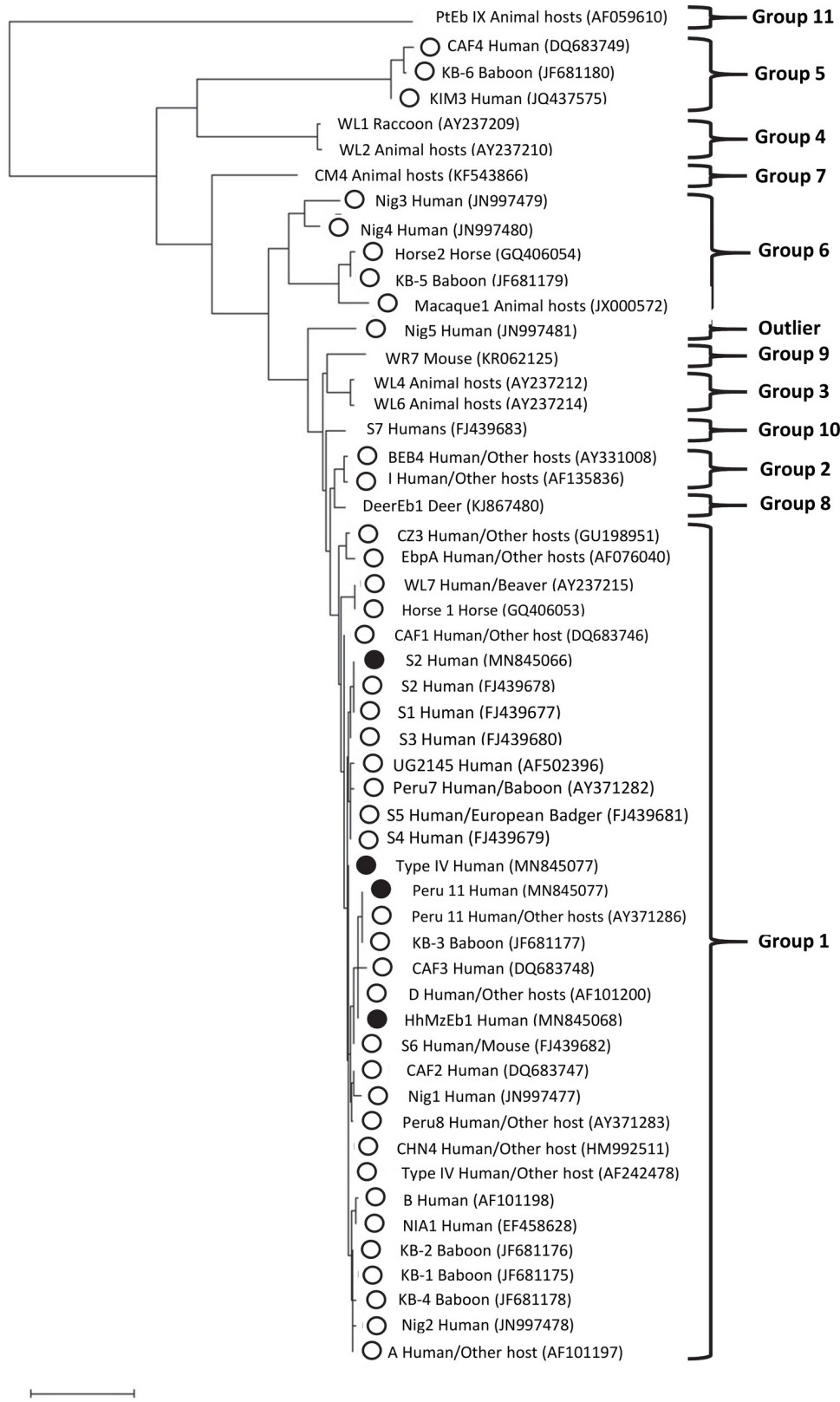

**Fig 1. Phylogenetic relationships among *Enterocytozoon bieneusi* genotypes identified in this study.** All genotypes identified in humans and animals in Africa, and genotypes to cover all groups of *E. bieneusi* were included for comparative purposes. Analyses were inferred by a Neighbor-Joining method of the entire ITS of rRNA gene based on genetic distances calculated by the Kimura two-parameter model (MEGA X software). All nucleotide sequences include host information with the GenBank accession number in parenthesis. Nucleotide sequences determined in this study are identified with black circles before the genotype name. White circles indicate genotypes identified in Africa.

missing values for water source and animal livestock in the Maputo dataset implied limited exploration of these associations, especially given the low numbers of cases.

Nucleotide sequence analysis on the ITS region of the nine positive cases revealed the presence of four genotypes, three previously reported (Peru11, Type IV, and S2) and a novel genotype (named HhMzEb1). This constitutes the first report of these genotypes in Mozambique. Genotypes Type IV and S2 were both identified in Zambézia and Maputo, while Peru11 was only identified in Zambézia and novel genotype HhMzEb1 only in Maputo. All genotypes identified in the present study, including novel HhMzEb1, belonged to Group 1 (Fig 1) that includes *E. bieneusi* genotypes with low host specificity that are found not only in humans but also in domestic and wild animals worldwide [10,11,14]. This is the second report of genotype S2 in humans, that had only been previously reported in Malawi from three children (2 HIV-positive and one HIV-negative) and eight HIV-positive adults [35]. Genotype Type IV (also reported previously as K, Peru2, BEB5, CMITS1, BEB-var, PtEB III) has widely been found in humans in different regions throughout the world [5,36–38], with multiple reports in African countries that include Gabon [36], Cameroon [36,39], Uganda [40], Niger [41], Nigeria [31,42,43], Democratic Republic of São Tomé and Principe [30], and Malawi [35] (S5 Table). In addition, Type IV has also been reported in a wide range of animal hosts including non-human primates, cattle, cats, dogs, rodents, birds, snakes, and black bears (S5 Table) as well as in water samples [44–47]. There are no previous reports of Peru11 (also reported previously as Peru12) in humans in Africa, but it has been found in humans in Asia and South America (S5 Table). In Africa, Peru11 has been found in a baboon in Kenya [48]. In addition, there are also reports of this genotype in other parts of the world in non-human primates, cats, raccoons, rabbits, rodents, and birds (S5 Table).

In this study, no mixed infections by different *E. bieneusi* genotypes were noticed using ITS-PCR and Sanger sequencing. However, it should be noted that co-infections seem a common finding in human populations in endemic areas. Indeed, intra-isolate diversity has been demonstrated in children with diarrhoea in Uganda using microsatellite markers (in addition to ITS) and subsequent cloning and sequencing of PCR products [49]. These data preclude us to conclusively state that mixed infections are not present in the surveyed Mozambican population as the methodology used in this study was not chosen to address mixed infections. The identification of potentially zoonotic genotypes Peru11 and Type IV, commonly found not only in humans but also in domestic and wild animals worldwide, indicates the potential for zoonotic or cross-species transmission. There are few studies in the African continent that include molecular characterization of *E. bieneusi* in humans and in animals. However, environmental (water) samples should be also considered as a source of infection by microsporidia, as *E. bieneusi* has been demonstrated to be involved in waterborne outbreaks of gastrointestinal disease [50]. Our findings emphasize the need of further studies to explore risk associations as these were limited in our study by the low prevalence and missing information. Studies could include water as well as animals that may be in contact with human populations in Mozambique to understand the modes of transmission of *E. bieneusi*, and, by analogy, of other diarrhoea-causing enteric protist species.

## Conclusions

This is the first molecular study of *E. bieneusi* in Mozambique. The parasite was primarily found in asymptomatic children in Maputo (Manhiça district) and Zambézia province. Molecular characterization detected a novel genotype (HhMzEb1) and three known genotypes (Type IV, Peru11 and S2) in diarrheal and healthy Mozambican children. The identification of genotypes previously described in animals suggests potential zoonotic and anthroponotic transmission. There is a lack of information of *E. bieneusi* in the African continent in humans, animal hosts and environmental samples. Under the One Health perspective, further genotyping studies are needed to better understand the epidemiology of this parasite.

## Supporting information

**S1 Fig. Occurrence of *Enterocytozoon bieneusi* in humans in Africa.** Average values for symptomatic and asymptomatic individuals are represented according to reported infection rates summarized in Table 1.
(TIF)

**S1 Table. Main socio-demographic features and risk factors of the asymptomatic schoolchildren population (*n* = 807) investigated in Zambézia province (Mozambique), 2017–2019.**
(DOCX)

**S2 Table. Main socio-demographic features and risk factors of the symptomatic schoolchildren population (*n* = 290) attended at public health centres in Zambézia province (Mozambique), 2017–2018.**
(DOCX)

**S3 Table. Main socio-demographic features and risk factors of the asymptomatic children population (*n* = 109) investigated in Maputo province (Mozambique), 2016–2018.**
(DOCX)

**S4 Table. Main socio-demographic features and risk factors of the symptomatic children population (*n* = 41) investigated in Maputo province (Mozambique), 2016–2018.**
(DOCX)

**S5 Table. Summary of all *Enterocytozoon bieneusi* genotypes reported in Africa including host and geographic range.** Highlighted in bold are African countries.
(DOCX)

## Acknowledgments

We thank the children and their caretakers who participated in the study, as well as the clinical, field and laboratory staff who worked tirelessly to ensure the data collection and laboratory testing was performed according to the protocols. We also thank all the local government authorities (district Administration Health and Education Directorates in Zambézia and Manhiça) and all community leaders for supporting the study.

## Author Contributions

**Conceptualization:** Inácio Mandomando, David Carmena.

**Data curation:** Inácio Mandomando, David Carmena.

**Formal analysis:** Aly S. Muadica, Augusto E. Messa, Jr., Alejandro Dashti, Sooria Balasegaram, Mónica Santin, Rafael Calero-Bernal.

**Funding acquisition:** Tacilta Nhampossa, Jason M. Mwenda, Inácio Mandomando, David Carmena.

**Investigation:** Aly S. Muadica, Augusto E. Messa, Jr., Alejandro Dashti, Filomena Manjate, Percina Chirinda, Marcelino Garrine, Delfino Vubil, Sozinho Acácio, Pamela C. Köster, Begoña Bailo, Tacilta Nhampossa.

**Methodology:** Sooria Balasegaram, Mónica Santin, Rafael Calero-Bernal, Inácio Mandomando.

**Project administration:** Percina Chirinda, Tacilta Nhampossa, Inácio Mandomando, David Carmena.

**Supervision:** Marcelino Garrine, Delfino Vubil, Sozinho Acácio, Tacilta Nhampossa, Inácio Mandomando, David Carmena.

**Validation:** Sooria Balasegaram, Mónica Santin, Rafael Calero-Bernal.

**Visualization:** Mónica Santin, Rafael Calero-Bernal, Inácio Mandomando, David Carmena.

**Writing – original draft:** Aly S. Muadica, Augusto E. Messa, Jr., Mónica Santin, Rafael Calero-Bernal, Jason M. Mwenda, Inácio Mandomando, David Carmena.

**Writing – review & editing:** Aly S. Muadica, Augusto E. Messa, Jr., Sooria Balasegaram, Mónica Santin, Rafael Calero-Bernal, Jason M. Mwenda, Inácio Mandomando, David Carmena.

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
