## [Decision Letter · Decision Letter 0]

13 Apr 2020

Dear Dr Carmena,

Thank you very much for submitting your manuscript "First identification of genotypes of Enterocytozoon bieneusi (Microsporidia) among symptomatic and asymptomatic children in Mozambique" for consideration at PLOS Neglected Tropical Diseases. As with all papers reviewed by the journal, your manuscript was reviewed by members of the editorial board and by several independent reviewers. In light of the reviews (below this email), we would like to invite the resubmission of a significantly-revised version that takes into account the reviewers' comments. 

We cannot make any decision about publication until we have seen the revised manuscript and your response to the reviewers' comments. Your revised manuscript is also likely to be sent to reviewers for further evaluation.

Sincerely,

Thuy Le

Guest Editor

Todd Reynolds

Deputy Editor

Reviewer's Responses to Questions

**Key Review Criteria Required for Acceptance?**

**Methods**

-Are the objectives of the study clearly articulated with a clear testable hypothesis stated?

-Is the study design appropriate to address the stated objectives?

-Is the population clearly described and appropriate for the hypothesis being tested?

-Is the sample size sufficient to ensure adequate power to address the hypothesis being tested?

-Were correct statistical analysis used to support conclusions?

-Are there concerns about ethical or regulatory requirements being met?

Reviewer #1: The manuscript by Salimo Muadica et al report on an extensive survey of Enterocytozoon bieneusi infection in children in Mozambique. These studies are difficult to conduct and the authors deserve credit for surveying a relatively large number of children.

Reviewer #2: (No Response)

Reviewer #3: -Are the objectives of the study clearly articulated with a clear testable hypothesis stated? YES

-Is the study design appropriate to address the stated objectives? YES

-Is the population clearly described and appropriate for the hypothesis being tested? YES

-Is the sample size sufficient to ensure adequate power to address the hypothesis being tested? YES

-Were correct statistical analysis used to support conclusions? YES

-Are there concerns about ethical or regulatory requirements being met? NO

"Comments to the Author":

In the epigraphs “Collection of stool samples in Zambézia province” and “Collection of stool samples in the Manhiça district, Maputo province”, in my opinion is confused the treatment and sample processing (lines 188-190 and lines 229-231, respectively). The authors could clarify or homogenize this step. 

- Line 243: “DNA extraction”. Add: “and purification”.

**Results**

-Does the analysis presented match the analysis plan?

-Are the results clearly and completely presented?

-Are the figures (Tables, Images) of sufficient quality for clarity?

Reviewer #1: Perhaps the most important contribution of the study is the apparent absence of association between diarrhea and Eb.

Reviewer #2: (No Response)

Reviewer #3: -Does the analysis presented match the analysis plan? YES

-Are the results clearly and completely presented? YES

-Are the figures (Tables, Images) of sufficient quality for clarity? YES

"Comments to the Author":

- Line 328: “Overall, E. bieneusi was detected by PCR”. Add: nested. 

- Table 1: Include in the table foot (legend), TEM (transmission electron microscopy).

- Table 3: Remove in table foot “NA: not applicable”, I think it is not used.

- S3 Table: Remove in table foot “NA: not applicable”, I think it is not used.

**Conclusions**

-Are the conclusions supported by the data presented?

-Are the limitations of analysis clearly described?

-Do the authors discuss how these data can be helpful to advance our understanding of the topic under study?

-Is public health relevance addressed?

Reviewer #1: Given the size of the study, the small number of positives is worth reporting, but statements comparing the prevalence in the 2 study sites (line 328, 412) and between this and previously published reports (lines 414-425) may not be warranted unless supported by a statistical analysis. Similarly, the statement on line 333 about the absence of mixed infections seems problematic given the PCR and sequencing method used in this study; see for instance Widmer G, Dilo J, Tumwine JK, Tzipori S, Akiyoshi DE. Appl Environ Microbiol. 2013, Sep;79(17):5357-62.

Reviewer #2: (No Response)

Reviewer #3: -Are the conclusions supported by the data presented? YES

-Are the limitations of analysis clearly described? YES

-Do the authors discuss how these data can be helpful to advance our understanding of the topic under study? YES

-Is public health relevance addressed? YES

"Comments to the Author": (Discussion).

- Lines 455-461: Comment: The authors should cite or comment the importance of the water transmission of the parasite for a better knowledge of the transmission pathways in the context in which the study was carried out.

**Editorial and Data Presentation Modifications?**

Reviewer #1: The manuscript is generally well written, but some additional editing seems warranted. See following examples:

Line 206-219. Consider reporting enrollment criteria as a numbered or bulleted list. Text using numerous parentheses and nested parentheses is difficult to follow.

Line 103. …capable of colonizing/infecting a broad spectrum…

Line 110. nearly 500 genotypes

Line 113. …mammalian species, including humans…

Line 126. …could be more frequent that anticipated. Despite the progress…

Line 156. delete “the” (…under reference number…)

Line 186. …seeking medical attention… (no “for” preposition)

Line 192. …children were between 3 and 14 years old.

Line 201. delete “continuous” (it’s implied in the sentence); something is missing after (HDDS), like program or survey?

Line 202. Start new sentence: This survey covers approximately…

Line 208. peripheral health facilities

Line 224. …were enrolled starting in April 2017.

Line 230. testing for enteric pathogens

Line 249. In Manhica, DNA was manually extracted and purified using the QIA….

etc.

Reviewer #2: (No Response)

Reviewer #3: "Comments to the Author":

Keywords: 

- Add: novel genotype.

Abstract:

- Line 33: “Enterocytozoon bieneusi (Phylum Microsporidia)”. Comment: With the new classification of the Eukaryotes, in my opinion the Microsporidia (Fungi) is not a Phylum. 

Reference: 1: Adl SM, Bass D, Lane CE, Lukeš J, Schoch CL, Smirnov A, Agatha S, Berney C,Brown MW, Burki F, Cárdenas P, Čepička I, Chistyakova L, Del Campo J, Dunthorn M,Edvardsen B, Eglit Y, Guillou L, Hampl V, Heiss AA, Hoppenrath M, James TY, Karnkowska A, Karpov S, Kim E, Kolisko M, Kudryavtsev A, Lahr DJG, Lara E, Le Gall L, Lynn DH, Mann DG, Massana R, Mitchell EAD, Morrow C, Park JS, Pawlowski JW, Powell MJ, Richter DJ, Rueckert S, Shadwick L, Shimano S, Spiegel FW, Torruella G, Youssef N, Zlatogursky V, Zhang Q. Revisions to the Classification, Nomenclature, and Diversity of Eukaryotes. J Eukaryot Microbiol. 2019 Jan;66(1):4-119. 

Introduction: 

- Line 92: “Microsporidia comprises 200 genera and nearly 1,500 species of spore-forming parasites”. Add: “Microsporidia (Fungi) comprises 200 genera and nearly 1,500 species of spore-forming parasites”.

- Lines 95-96: “causing gastrointestinal infections globally”. Add: “and other disorders and pathologies such as in lung”.

Reference: “del Águila C, Lopez-Velez R, Fenoy S, Turrientes C, Cobo J, Navajas R, Visvesvara GS, Croppo GP, Da Silva AJ, Pieniazek NJ. Identification of Enterocytozoon bieneusi spores in respiratory samples from an AIDS patient with a 2-year history of intestinal microsporidiosis. J Clin Microbiol. 1997, Jul; 35(7): 1862-6”. 

- Lines 105-107: “Infections are acquired through ingestion of food or water contaminated with E. bieneusi spores, or through direct contact with faeces of infected persons and animals”. Add: soils. 

- Lines 118-120: “GEMS included in their study enteric protozoan parasites Giardia duodenalis, Cryptosporidium spp., and Entamoeba histolytica, but not E. bieneusi”. Comment: E. bieneusi is not a protozoa parasite. Add: enteric parasites, for example. 

 - Line 130: “asymptomatic paediatric populations”. Change for “pediatric”. 

Materials and Methods: 

In the epigraphs “Collection of stool samples in Zambézia province” and “Collection of stool samples in the Manhiça district, Maputo province”, in my opinion is confused the treatment and sample processing (lines 188-190 and lines 229-231, respectively). The authors could clarify or homogenize this step. 

- Line 243: “DNA extraction”. Add: “and purification”. 

Results:

- Line 328: “Overall, E. bieneusi was detected by PCR”. Add: nested. 

Discussion: 

- Lines 455-461: Comment: The authors should cite or comment the importance of the water transmission of the parasite for a better knowledge of the transmission pathways in the context in which the study was carried out. 

Tables:

- Table 1: Include in the table foot (legend), TEM (transmission electron microscopy).

- Table 3: Remove in table foot “NA: not applicable”, I think it is not used.

- S3 Table: Remove in table foot “NA: not applicable”, I think it is not used.

References:

- Add: “Adl SM, Bass D, Lane CE, Lukeš J, Schoch CL, Smirnov A, Agatha S, Berney C,Brown MW, Burki F, Cárdenas P, Čepička I, Chistyakova L, Del Campo J, Dunthorn M,Edvardsen B, Eglit Y, Guillou L, Hampl V, Heiss AA, Hoppenrath M, James TY, Karnkowska A, Karpov S, Kim E, Kolisko M, Kudryavtsev A, Lahr DJG, Lara E, Le Gall L, Lynn DH, Mann DG, Massana R, Mitchell EAD, Morrow C, Park JS, Pawlowski JW, Powell MJ, Richter DJ, Rueckert S, Shadwick L, Shimano S, Spiegel FW, Torruella G, Youssef N, Zlatogursky V, Zhang Q. Revisions to the Classification, Nomenclature, and Diversity of Eukaryotes. J Eukaryot Microbiol. 2019 Jan;66(1):4-119”. 

- Add: “del Águila C, Lopez-Velez R, Fenoy S, Turrientes C, Cobo J, Navajas R, Visvesvara GS, Croppo GP, Da Silva AJ, Pieniazek NJ. Identification of Enterocytozoon bieneusi spores in respiratory samples from an AIDS patient with a 2-year history of intestinal microsporidiosis. J Clin Microbiol. 1997, Jul; 35(7): 1862-6”.

**Summary and General Comments**

Reviewer #1: The apparent absence of association between diarrhea and Eb is worth reporting. Some of the conclusions should be supported by statistical tests.

Reviewer #2: (No Response)

Reviewer #3: The main objective of the manuscript is very interesting with promising results. In my opinion would have been interesting to investigate the presence of other species of Microsporidia (for example Encephalitozoon spp) using others complementary techniques such as staining methods or immunological techniques.

Obviously, the authors want to study only the genotypes of E. bieneusi in children in Africa (Mozambique) as main objective. But these additional techniques (staining methods or immunological techniques) could contribute to knowledge of the prevalence of other species of Microsporidia and provide us of a valuable information for this study or for new studies in the future. 

I enjoyed reviewing this article. It is extensive with detailed information with a complete review of the prevalence of E. bieneusi and its genotypes in Africa. The results are interesting highlighting the description of a novel genotype. I encourage the authors to continue investigating the circulation and prevalence of this novel genotype and those already described in Africa. 

PLOS authors have the option to publish the peer review history of their article (what does this mean?). If published, this will include your full peer review and any attached files.

Reviewer #1: Yes: Giovanni Widmer

Reviewer #2: No

Reviewer #3: Yes: Dr. Fernando Izquierdo Arias.
---

## [Editor Report · Decision Letter 1]

23 May 2020

Dear Dr Carmena,

We are pleased to inform you that your manuscript 'First identification of genotypes of Enterocytozoon bieneusi (Microsporidia) among symptomatic and asymptomatic children in Mozambique' has been provisionally accepted for publication in PLOS Neglected Tropical Diseases.

Best regards,

Thuy Le

Guest Editor

Todd Reynolds

Deputy Editor

---

## [Editor Report · Acceptance letter]

18 Jun 2020

Dear Dr Carmena,

We are delighted to inform you that your manuscript, "First identification of genotypes of Enterocytozoon bieneusi (Microsporidia) among symptomatic and asymptomatic children in Mozambique," has been formally accepted for publication in PLOS Neglected Tropical Diseases.

Best regards,

Shaden Kamhawi

co-Editor-in-Chief

Paul Brindley

co-Editor-in-Chief
